# A Cylindrical Optical-Space Black Hole Induced from High-Pressure Acoustics in a Dense Fluid

Edward A. Rietman [1,2,*], Brandon Melcher [2] , Alexey Bobrick [2,3] and Gianni Martire [2]

1    Manning College of Information and Computer Science, University of Massachusetts, Amherst, MA 01002, USA
2    Applied Physics, 477 Madison Avenue, New York, NY 10022, USA
3    Technion-Israel Institute of Technology, Physics Department, Haifa 32000, Israel
*    Correspondence: ed@appliedphysics.org

**Abstract:** We describe the construction of an optical-space, cylindrical black hole induced by high pressure in a dense fluid. Using an approximate analogy between curved spacetime and optics in moving dielectric media, we derive the mass of the black hole thus created. We describe the resulting optical-space using a Bessel beam profile and Snell's law to understand how total internal reflection produces a cylindrical, optic black hole.

**Keywords:** GRIN; photonic; lead zirconate titanate; PZT; black hole

## 1. Brief Review and Background

### 1.1. Metamaterials

Since much of the recent and current research exploits the use of metamaterials for analog simulation of gravity and general relativity in the lab, we would be remiss not to briefly review it. Metamaterials are two- and three-dimensional periodic arrangements of matter. This periodic arrangement is known as a "crystal" and the "atoms" comprising the crystal range in size from submicron to centimeters. The size of the atom dictates the wavelength of the "light" that is modulated upon interaction with the crystal. If the crystal is comprised of colloidal size particles it is known as a photonic crystal. Of course, larger metallic "atoms", even tiny inductor circuits, can be arrayed into a two-dimensional pattern, making that crystal into a photonic crystal in microwave or radio frequency. If the atoms are, for example, marbles, the crystal will modulate acoustic waves. It is then known as a phononic crystal.

The first comprehensive introduction to metamaterials was by [1] and focused on photonic crystals. The field of photonic and phononic crystals quickly took off and we will not attempt to review the literature. Recent reviews are given by [2,3]. A few years ago, we conducted a computational survey of two-dimensional phononic crystals for band-gap engineering [4]. Another review of acoustic metamaterials for sound focusing and sound confinement can be found in [5]. In prior laboratory work, we described experimental work using ultrasonics in liquid lenses for tunable optical devices [6]. A particularly interesting recent monograph on the "new science of sound" is given by [7].

### 1.2. Analog Astrophysical Models

We now review some of the fascinating work that has been reported on creating, in the laboratory, astrophysical analogs. An edited book [8] reviewed artificial black holes. The systems covered include Bose–Einstein condensates—a phenomena that occurs at liquid helium temperature, and novel condensed matter phenomena that emulate black holes and quantum gravity. More recently, [9] have described virtual magnetic monopoles and wormholes in electromagnetic metamaterials. Their work is analogous to electromagnetic

and acoustic cloaking. We do not review cloaking applications and focus only on astrophysical models. An extensive review of analog gravity is given by [10], and we refer the reader interested in other examples of analog systems and their quantum gravitational applications to that text and the citations within.

Researchers have described experimental electromagnetic black holes at frequencies of $\mathcal{O}(10)$ GHz [11]. Their device consisted of split-ring dielectrics, a common approach to electromagnetic metamaterials. On a much smaller scale, in the optical regime, ref. [12] proposes and conducted extensive simulations of optical black holes in photonic crystals. This is valid because, as pointed out by [13] the language of classical optics and Riemannian geometry for gravity is isomorphic.

The authors of [14] used ultra-short laser pulses to affect the microstructure of optical fibers; in other words, to modulate the refractive index. These modulations, or perturbations, traveled down the optical fiber and an infrared laser pulse was used as a probe. They measured a 13 nm blue-shift at a white hole event horizon. In similar work on optical fibers, refs. [15–18] use a graded index optical fiber with Bessel function refractive index. They measured Hawking radiation.

Smolyaninov and his colleagues have done extensive work, both in computational simulation and laboratory work, in studying artificial astrophysical and cosmological systems. Here we will briefly review some of that work. Two papers [19,20] describe how optical space in metamaterials may be engineered to mimic the landscape of a multidimensional universe. The nonlinear optics in these materials emulates the symmetries of Kaluza–Klein theories, including particle creation showing flashes of light reminiscent of singularities at the birth of universes. Smolyaninov has emulated virtual black hole creation by optically observing the thermal phase transition of a mixture of aniline and cyclohexane just below the critical temperature [19]. In another paper [21] they describe the modeling of time and big bang emulation in microfabricated plasmonic hyperbolic metamaterials. They show that in the microdevice multiple big bangs occur producing bursts of light. In a similar microfabricated device [22] observed simulated braneworld collisions and creations of 4-D spacetimes. Rather than construct the microdevice using lithography and etching, Smolyaninov's team used a ferrofluid and a magnetic field to align the fluidic particles to form a metamaterial for modulation of light at 1500 nm [23]. Again, they observe multiuniverse formation similar to phase changes.

In much of the above, Smolyaninov and his colleagues investigated big bangs and meta-universe formation, but these model systems lack causality, or world lines. In [22], Smolyaninov describes an electromagnetic metamaterial that demonstrates causality. Specifically, it involves modulation of electric dipoles in the metamaterial allowing signals to propagate in only one direction.

Since acoustic waves in a medium result in a densification of the media, it is reasonable to assume that there is a small gravitational mass carried by sound waves. Theoretical results showing that sound waves may be affected by gravity, and that the traveling sound waves generate tiny gravitational fields appeared in [24]. Gravity wave antennae built from engineered metamaterials were discussed in [25]. The authors of [26] then show finite element modeling of interaction of surface gravity waves with a mechanical metamaterial induced by periodic underwater oscillating resonators. Metamaterials have been proposed as potential Casimir effect and gravitational wave detectors [27]. Their response to visible light was measured in [28]. Lastly, a split ring resonator metamaterial model of the Alcubierre warp drive that is capable of accelerating up to 1/4c was described in [19].

*1.3. Introduction to Gordon's Metric*

An important observation for the analog models above is that a variable speed of light in a medium can be alternatively phrased in the terminology of curved spacetimes [13,29,30]. In pursuit of this analogy, Novello and Bittencourt [31] review the Gordon metric, which describes electromagnetic waves in a moving medium with an

isotropic dielectric constant. The electromagnetic waves propagate as geodesics not in the usual background geometry, but follow the optical metric:

$$\bar{g}_{ab} = g_{ab} + \left(1 - \frac{1}{\mu\epsilon}\right)u_a u_b, \tag{1}$$

$$\bar{g}^{ab} = g^{ab} + \left(1 - \mu\epsilon\right)u^a u^b. \tag{2}$$

The barred quantities correspond to those in the optical equivalent spacetime. We can now write the relationship between the two determinants as:

$$\det \bar{g}_{ab} = \frac{1}{n(x)^2}\det g_{ab}, \tag{3}$$

where we have defined the refractive index $n(x) = \sqrt{\epsilon\mu}$.

This enables us to explore new theoretical and experimental opportunities with curved spacetime in the lab. For example, with respect to new theoretical avenues, Chen and Kantowski [32–34], have used the Gordon metric to describe a cosmological dark refractive index by incorporating both apparent size and luminosity redshifts comprising optical absorption.

It was shown in [13] that, from an optics perspective, the above metric for a stationary ($u_a = \delta_a^0$), non-homogeneous dielectric ($\mu\epsilon = n^2 = n(\mathbf{x})^2$) is equivalent to the Schwarzschild metric when

$$n^2 = \left(1 + \frac{M}{2r}\right)^6 \left(1 - \frac{M}{2r}\right)^{-2}, \tag{4}$$

with $M$ the central mass, and $r$ is a radial coordinate.

Other computational examples are given by, Wang and Chen [35] and, Narimanov and Kildishev [36]. Both sets of authors [35,36] describe an approach to numerical simulation of a graded-index photonic crystal, which they demonstrate is an optical-space black hole. Their photonic crystal would be built from an appropriately arranged array of dielectric cylinders and produce a cylindrical-shaped optical black hole. More accurately, their photonic crystal is known as a graded-index (GRIN) lens which typically produces a Bessel beam profile [37]. These Bessel beam profile lenses are also known as axicon lenses [38]. Higginson et al. [6] used a cylindrical tube-shaped piezoelectric transducer and measured the profile of an induced GRIN lens. Here we extend that work and demonstrate that an optical black hole can be induced from high-pressure acoustics in a dense fluid. The pressure induced in the fluid is large enough (200 MPa, measured with a needle hydrophone) to result in total internal reflection.

In the following, we first describe our experimental approach and the results. Then, we describe our analysis of the results and finally we conclude with some potential applications of GRIN lenses and this approach to modeling optical-space black holes.

## 2. Materials and Methods

When high frequency signals are applied to a tube-shaped acoustic transducer they can induce very high-pressure standing waves, especially if the fluid in the tube is dense. We used a PZT (lead zirconate titanate) ceramic piezoelectric tube transducer from Stemic (Davenport, FL, USA), part number SMC4037T50111, which is 40 mm in O.D, 50 mm in length and has a key resonance at 36 kHz. In order to deliver more power to the piezo tube, we operated the tube at the second harmonic (i.e., 72 kHz) and amplified a 20Vpp (volts peak to peak) square wave to 74Vpp with little distortion in the delivered pulse (other than the expected ringing). The tube transducer is shown in the inset photo of Figure 1. Our laser is a Thorlabs HRS015 HeNe laser. The fiber collimator is a Thorlabs F230SMA–1550, and the spectrometer is a Thorlabs CSS 100 series. Figure 1 shows the optical setup.

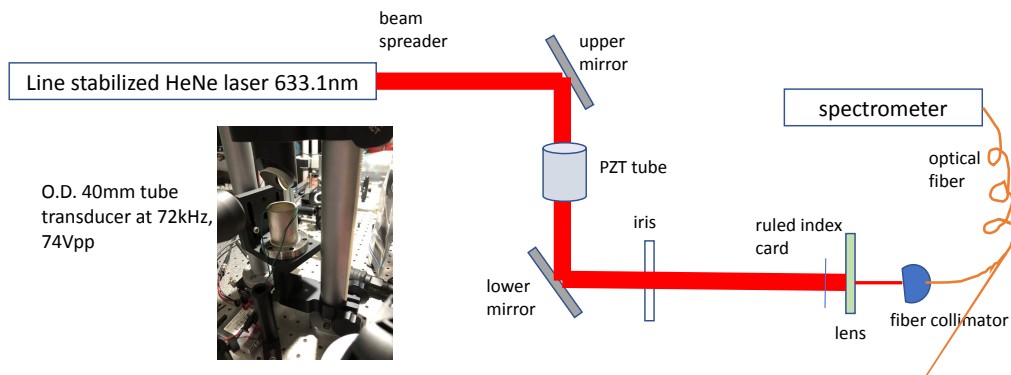

**Figure 1.** The laser beam passes through a beam spreader, then via a mirror through the PZT tube, another mirror, an iris, and finally either to a ruled index card or focused into the fiber collimator, where the light is analyzed by the spectrometer. The inset photo shows the PZT tube mounted between the two beam steering mirrors. The PZT tube was filled with pharmaceutical grade glycerin.

## 3. Results

Because we wanted to analyze the light from the tube device with the spectrometer, we focused the light with a lens into the fiber collimator. We also wanted to make some photographs of the dynamics taking place in the tube, we used a ruled index card to project the light. The experiment was done in three parts: (1) spectrometer light capture in forward direction, i.e., light through PZT tube; (2) then spectrometer light capture in the reverse direction (configuration not shown in Figure 1) to capture a light backwards through the device; and (3) with two Thorlabs NDUV10A neutral density (ND) filters in the optical path, just before the beam spreader, to reduce the intensity so as to capture the projected image on a centimeter-ruled index card in the forward direction. The ND filters were used to prevent saturation of the webcam pixels. Figure 2 shows the results from a measurement of the light captured in the spectrometer in the forward direction. The first thing to notice is the red HeNe trace with peak intensity of 1.00 and peak wavelength 633.1 nm; also notice the two vertical dashed marker lines. These are placed at the edges of the lowest point on the HeNe line (632.8 nm and 634.0 nm). As shown in the inset photo, when the power is not turned on to the PZT tube, we obtain an almost uniform disk of light corresponding to the HeNe line, the red trace in the spectrum. However, when we apply the power to the PZT tube (the green trace in the spectrum) at first there is an increase in intensity and line broadening (including a slight red shift), but that is not stable and within a second a black region quickly opens up and grows until it fills about 1/4 of the area (at 2 s). At this point the remaining light is too low in intensity to be measured by the spectrometer, as shown by the blue trace in the spectrum.

We would like to emphasize that the bright rings observed in the PZT are not stable. The number of rings was variable across experimental runs. They appear to depend upon more than the pressure and refractive index.

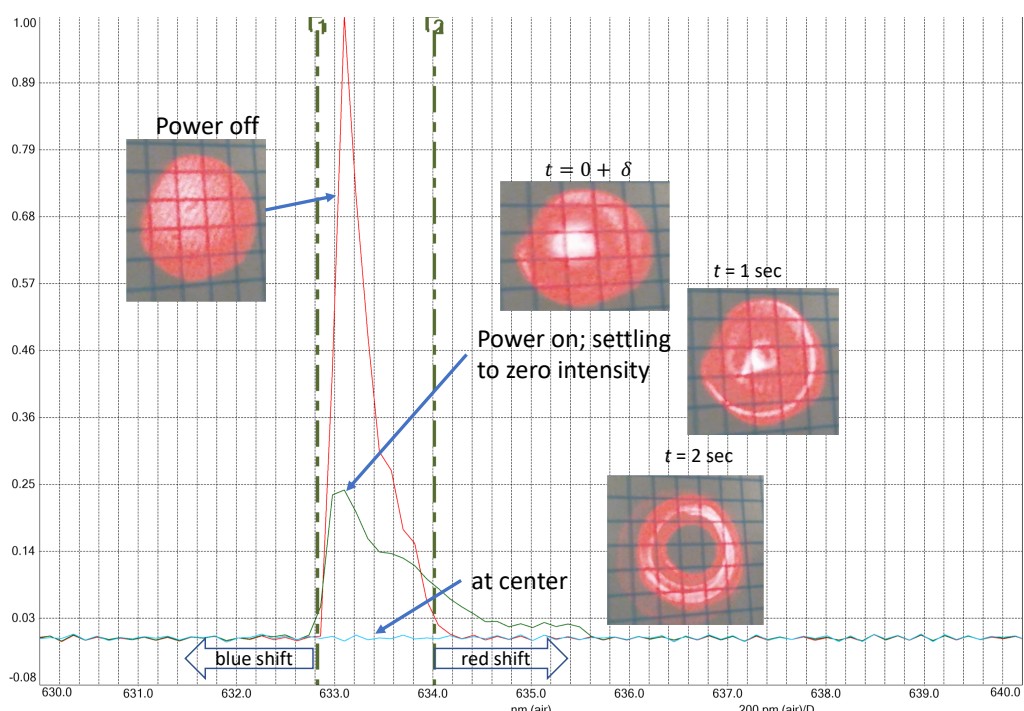

**Figure 2.** Results from the spectrometer. See text for details.

## 4. Discussion

To develop an intuition for this phenomenon we now discuss several explanations: Snell's law and Bessel beam profile, Schwarzschild radius, refractive index from pressure measurements, and a comparison with optical fiber.

### 4.1. Snell's Law and Bessel Beam Profile

We will be referring to Figures 3 and 4 to describe how it is optically possible to create an analog cylindrical black hole. Figure 3 shows the initial instant power was applied to the PZT tube ($t = 0$ s). During the time from $t = 0 + \delta$ s to $t = 1$ s, the light is being focused to an intense optical singularity (where $\delta$ is a very small increment of a second, on the order of 10 s of ms). This is caused by a moving front in the dense fluid, which modulates the refractive index causing the intensity change. At time $t = 1$ s we can see the black hole is beginning to open up. This is the critical angle of 0.74387 radians. During the time between $t = 1$ s and $t = 2$ s, more and more light is totally reflected. After $t = 2$ s, the cylindrical black hole is stable. However, we typically turn off the power at about $t = 15$ s to avoid heating of the fluid from the power delivered to the PZT tube. If we continue powering up the transducer, the black hole becomes unstable and the fluid exhibits, from an optical perspective, multiple-moving optical singularities, due to thermal currents being setup in the tube.

As a demonstration of the Bessel beam profile, consider Figure 4. The equation for the pressure of an acoustic standing wave in a cylindrical cavity as given by [6,39]:

$$P(r,t) = A J_0(kr)\cos(\omega t), \tag{5}$$

where $A$ is the amplitude of the pressure wave, set to 200 MPa (as measured by a needle hydrophone), $J_0$ is the Bessel function of the first kind, $r$ is radius (20 mm), $k$ is the wave number, $\omega$ is the angular frequency (72 kHz), and $t$ is the time shown in the plot in Figure 4b. The plot on the left of Figure 4 is simply the Bessel function Equation (5) overlaid on the photo of the light transmitted through the PZT tube. It is not an exact fit. It is important to realize that not all the light rings are necessarily visible from the Logitech web camera. We show it only as a guide to understanding the shape of the cylindrical black hole with surrounding shells in optical-space. Furthermore, the 2D projection shown on the left of

Figure 4 is a result of internal light scattering. Ideally, if we could get an accurate 3D image from a time instant, what we would see inside the PZT tube would be concentric tubes of refraction index modulations. The light rings in the projection are the regions of lower refractive index, and the black hole and the dark rings are the peaks and troughs in the pressure-time-radius 3D plot shown in Figure 4b. The stability of the cylindrical black hole is likely due to the viscosity of the fluid (648 cP) [40], which can't respond at the speed of sound in the fluid (1920 m/s) [41].

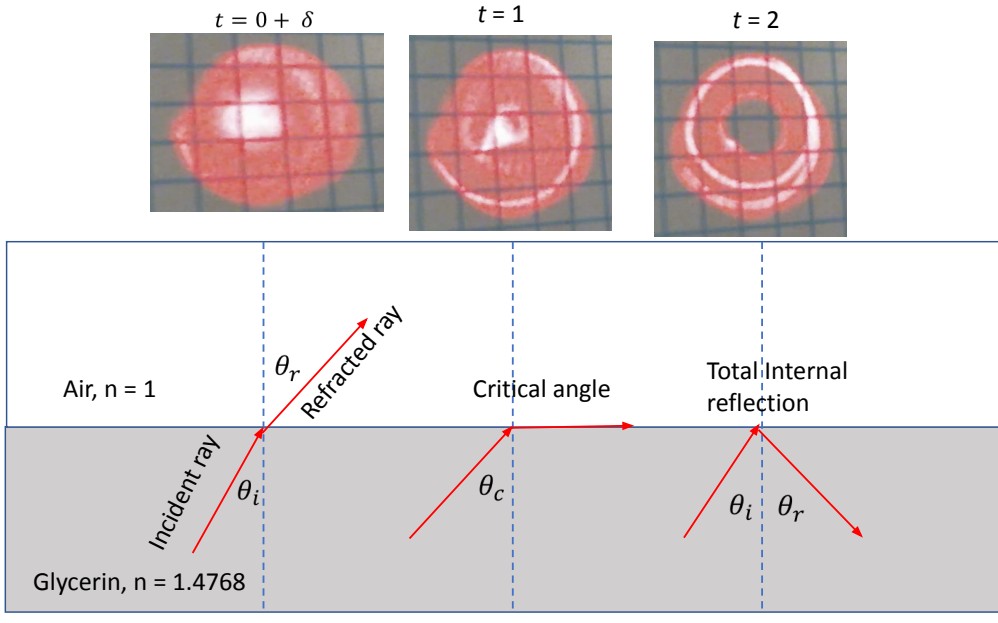

$$\theta_c = \arcsin(1/1.4768) = 0.74387 \text{ radians}$$

**Figure 3.** A possible explanation of the pressure-induced analog black hole. See [42] for refractive index study as function of pressure.

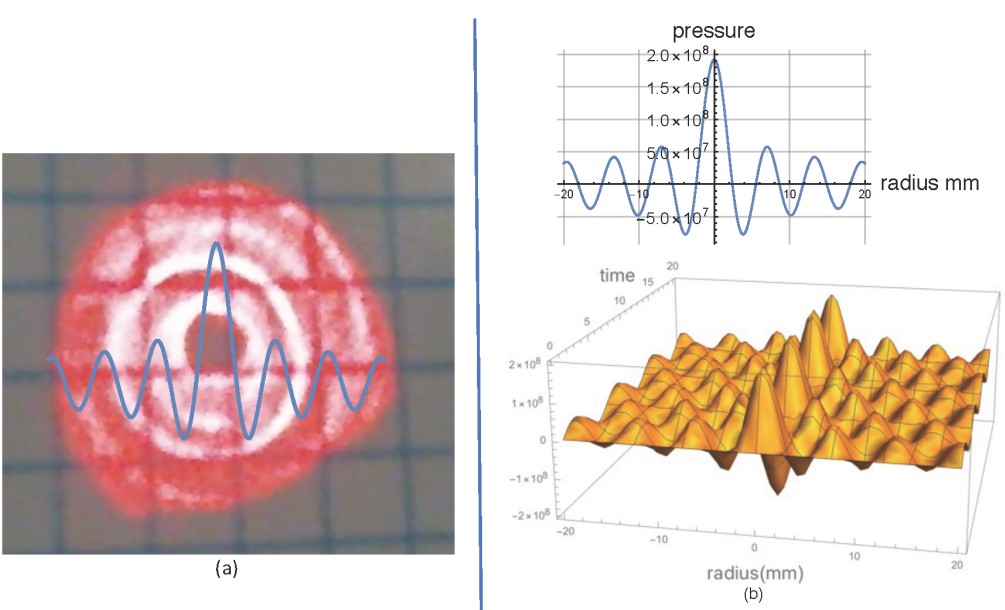

**Figure 4.** A Bessel function fit of image of the analog black hole. Panel (**a**) shows the fit to the image, and panel (**b**) shows the 3D radius, pressure, and time plot.

Above, we suggested that the cylindrical black hole has total internal reflection. That is not exactly correct, but almost. The critical angle at the glycerin–air boundary is 0.74387 radians (about 42.6 degrees) and the critical angle at the bottom of the device (quartz,

$n = 1.5525$) is 0.6998 radians (40.1 degrees). By using a beam splitter (not shown in Figure 1) before the upper mirror, we can capture the backward light onto a screen or detector. Though the dim light can be seen with the naked eye on the screen in a totally darkened room, it was too dim to register on the spectrometer or the CCD camera. In addition to scattering off the walls, there could be scattering from individual and multiple 3D cylindrical shells of refractive index modulations that would also give rise to internal reflection. Therefore, we essentially have total internal reflection.

### 4.2. Refractive Index from Pressure Measurement

We now turn our attention to the refractive index of the black hole (the core) and the surrounding region (the shell). In this section we will also compare the empirical and theoretical results. Since we know the refractive index of the outer region of the black hole, the shell, is $n = 1.4768$, and since we measured the inner core pressure to be about 200 MPa, we can estimate the refractive index in the core region. Using the Tait equation [43] (a relationship between pressure and density of liquids), Higginson et al. [6] derived the time-invariant relationship for density $\rho_2$:

$$\rho_2 = \langle \rho - \rho_0 \rangle = \frac{2 - \gamma}{2\rho_0 c_0^4} \langle p^2 \rangle - \frac{\rho_0}{2c_0^2} \langle u \cdot u \rangle. \tag{6}$$

In this equation, $\langle \rho - \rho_0 \rangle$ is the time average density; $\rho_0$ is the density of the liquid (glycerin), 1260 kg/m³; $c_0$ is the speed of sound in the liquid under standard conditions, 1920 m/s; $\gamma$ is an empirical (unitless) constant set to 10.0 [6]; $u$ is the instantaneous velocity vector for the pressure wave; and $p$ is the pressure of the acoustic wave. Following [6] for circularly symmetric acoustic standing wave, we substitute Equation (4) into Equation (8) and obtain:

$$\rho_2 = \frac{A^2}{4\rho_0 c_0^4} \left( (2 - \gamma) J_0^2(kr) - J_1^2(kr) \right), \tag{7}$$

where $A$ is the sound pressure amplitude ( 200 MPa), $J_i$ is a Bessel function of the first kind, and $r$ is the radial coordinate. Now we use the Lorentz–Lorenz relation [6,44] to get the time-averaged refractive index:

$$\langle n \rangle = n_0 - \alpha \left( (2 - \gamma) J_0^2(kr) - J_1^2(kr) \right), \tag{8}$$

with

$$\alpha = \frac{QA^2(n_0^2 + 2)^2}{24\rho_0 c_0^4 n_0}, \tag{9}$$

where $Q$ is molar refractivity (for glycerin $2.23 \times 10^{-4}$ m³/kg), $n_0 = 1.4768$ is the refractive index at $\rho_0$. Substituting the various values for glycerin, and with $\gamma = 10.0$, we get $n = 1.4784$ at 200 MPa. See Appendix A for a discussion of the validity of the above formulas over a range of applied pressures.

### 4.3. Schwarzschild Comparison and Optical Mass

As discussed above, the Schwarzschild black hole metric can be cast in coordinates such that, from the point of view of electromagnetism, it is equivalent to a stationary dielectric medium with radius-varying refractive index. In (4), we have the exact correspondence. In the limit $2r \gg M$, this refractive index has the form

$$n^2 = 1 + \frac{4M}{r} + \mathcal{O}\left( \frac{1}{r^2} \right). \tag{10}$$

In a similar vein, we can use the asymptotic form for the Bessel functions:

$$\lim_{x \to \infty} J_n(x) = \sqrt{\frac{2}{\pi x}} \cos\left( \frac{\pi}{4}(1 + 2n) - x \right) \tag{11}$$

We use this in (8) and take the square to obtain

$$\langle n \rangle^2 \simeq n_0^2 + \frac{18 n_0 \alpha}{\pi k r}. \tag{12}$$

We can always rescale the time coordinate by a constant in General Relativity, and we use $t \to n_0 t'$ to match (4). This means the metric we identify as equivalent to our experimental set-up is

$$n^2 = 1 + \frac{18\alpha}{\pi k r n_0}. \tag{13}$$

This means we can identify our glycerin medium of differential pressure with a black hole of mass

$$\begin{aligned} \frac{G}{c^2} M &= \frac{3}{16\pi k} \frac{Q A^2 (n_0^2 + 2)^2}{\rho_0 c_0^4 n_0^2} \\ &= 6.8 \times 10^{-6} \text{ m,} \end{aligned} \tag{14}$$

where we restored the correct units to the lefthand side. Dividing through by those factors

$$M = 9.1 \times 10^{21} \text{ kg} \simeq 10^{-3} M_\oplus \tag{15}$$

The astute reader will notice that the matching we performed in this subsection does not quite mesh. We took the Schwarzschild solution with spherical symmetry and matched it to a Gordon metric associated with a cylindrical dielectric. The authors are unaware, as of publication, of cylindrical solutions in General Relativity that allow the same procedure of matching to the Gordon metric.

That being said, we find that the error between the exact and approximate Schwarzschild refractive index, Equation (10), is less than ten percent at $r \gtrsim M = R_s/2$. Therefore, we can trust this approximation holds to radii below the Schwarzschild radius. In addition, the presence of a dark spot and the related redshift further cement the trust in this analogy.

While the black hole analogy we have created has a quite small mass, the associated Hawking radiation is too small to impact our set-up. The temperature of a solar mass black hole is $10^{-8}$ Kelvin. The corresponding temperature for our set-up is only $\mathcal{O}(10)$ Kelvin, which is too cold to create any observable effects.

Creating an analogous system to a regular black hole (see, e.g., [45]) is an interesting possibility, as one can clearly not construct singularities in real materials. This would involve more control over the pressure profile within the glycerin (or other fluid). It would also only apply to that given model of a regular black hole; thus, it may not be generalizable.

### 4.4. Comparison of Cylindrical Black Hole with Fiber Optics

Our estimate for refractive index in the core region of the cylindrical optical black hole is $1.4770 < n_{core} < 1.4784$ and the shell value is $n_{shell} = 1.4768$. For comparison, [36] reports on an optical black hole in a glass fiber. Optical fibers can be made with a higher refractive index core. In their simulations, they let the outer region of the fiber have a permittivity of $\epsilon_0 = 2.1$ and the doped core is $\epsilon_c = 12 + 0.7i$. Since we know that $n = \epsilon \mu$, we can calculate the shell and core refractive index. Using free space $\mu = 1.0$, for the shell of the optical fiber we get $n_{shell} = \sqrt{2.1} = 1.449$. The actual value for fused silica is 1.444 [46]. For the core of the optical fiber [36] reports $n_{core} \cong \sqrt{12} = 3.46$. The typical core refractive index is 1.447 [46]. We have not included the imaginary part $\epsilon_c = 12 + 0.7i$, which would be related to the doping of the fiber. For telecommunications optical fiber we can see that the shell refractive index differs from the core by about $\delta n = 0.003$, which is sufficient to result in total internal reflection. Thus, it is easily conceivable that our system can result in total internal reflection. At the low-end estimate for our system we get $\delta n = 0.0002$ and at the high-end we get $\delta n = 0.0016$.

## 5. Conclusions

We have demonstrated an easily constructed optical-space cylindrical black hole that is induced from high-pressure in a dense fluid. We have also shown how to interpret the results from a variety of perspectives Taking a lead from [36] we compared our cylindrical black hole to fiber optics and lastly, we showed how the refractive index would change the speed of light in a background media with some near real-world group velocities and some relativistic velocities.

With respect to potential applications, [6,35] mentioned a GRIN lens or axicon lens, and [36] speculates that the optical black hole they analyzed may have applications in broadband omnidirectional light absorbers (that would only be true within a certain range of frequencies). They go on to further speculate that such a perfect absorber may have applications in optoelectronic circuits, and solar light harvesting, though they don't give any examples. We also want to point out that [36] uses the analogy of an optical fiber to describe their cylindrical black hole. It is well known in telecommunications that optical fibers have a higher core refractive index than the shell of the fiber [46]. This allows the optical signal to travel great lengths without dispersing.

We find the application suggested by [47] to be very original. They demonstrated that a piezoelectric tube filled with a dense gas (e.g., Ar, SF6) could guide a high-energy laser beam, in the same way an optical fiber guides a light pulse in telecommunications. We speculate that it might be useful in focusing very high-power lasers for various applications including laser drilling and laser-induced fusion.

Of course, the main purpose of studying these analog astrophysical phenomena are to inform and guide studies of quantum gravity and quantum cosmology. To this end both computational analogs and lab bench-scale analogs could, we envision, contribute to this larger goal and eventually to novel quantum technologies [48].

**Author Contributions:** E.A.R. designed and conducted the experimental work. E.A.R. wrote the Experimental, Results section and the first draft of the Discussion. B.M. and A.B. assisted in analyzing the data, and writing the paper. G.M. oversaw the project and funding acquisition. All authors have read and agreed to the published version of the manuscript.

**Funding:** This research received no external funding.

**Institutional Review Board Statement:** Not applicable.

**Informed Consent Statement:** Not applicable.

**Data Availability Statement:** The authors are happy to share the data with interested parties. Please email the corresponding author for access.

**Acknowledgments:** The project was funded by Applied Physics Inc, New York. EAR Thanks Keith Higginson and Bart Lipkens for helpful discussions. We also thank Jae-Hyeon Ko [42] for use of the data in Figures A2 and A3.

**Conflicts of Interest:** The authors declare no conflict of interest.

## Appendix A

In Section 4.3 we described the relationship between refractive index and pressure. The plot for the time average of the refractive index Equation (8) is shown in Figure A1 and implies that the refractive index would follow a second-order growth curve. This is not realistic for real-world materials. Jeong et al. [42], using a technique known as Brillouin spectroscopy measured refractive index change for glycerin from 0 to 30 GPa. Their data (reproduced with permission) is shown in Figures A2 and A3. The first of the two shows a second order fit of pressure in GPa to refractive index. The latter shows a linear fit, with standard error bars, of the first four data points from the larger plot. Interpolating on the linear fit we get 1.4770 for the refractive index at 200 MPa.

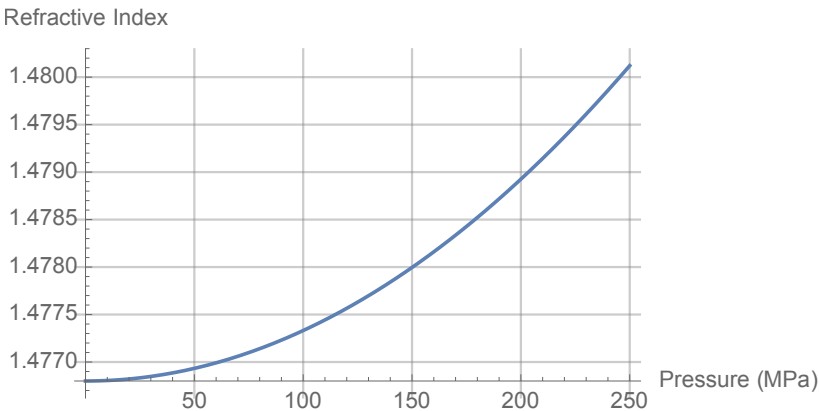

**Figure A1.** A plot of (8). See text for details. At 200 MPa the refractive index is 1.4784.

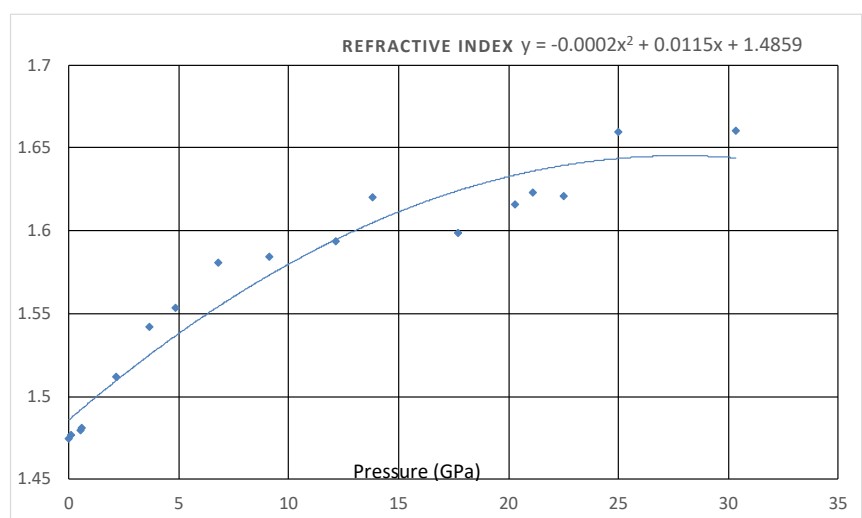

**Figure A2.** Data from Jae-Hyeon Ko of [42]; used with their permission. See text for discussion.

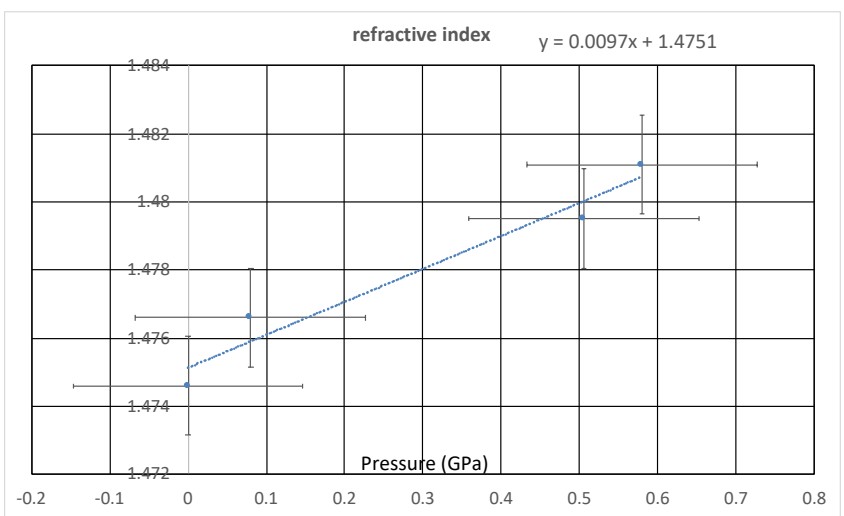

**Figure A3.** Data from Jae-Hyeon Ko of [42]; used with their permission. See text for discussion.

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
