# Peer review of "A Cylindrical Optical-Space Black Hole Induced from High-Pressure Acoustics in a Dense Fluid"

_universe, doi:10.3390/universe9040162_

Round 1

Reviewer 1 Report

The word "analog" should appear in the title.

More problematic is how the analogy is carried out: the exact equation (4) is approximated by (10) only at very large r >> M/2, while the black hole's horizon is of course at r = 2M, which doesn't seem to satisfy the criterion. How useful then is the analogy? Should Hawking radiation, for example, be expected to be similar? (of course the authors themselves note that their geometry is cylindrical, while there are no cylindrical black holes in 3+1 spacetime...)

Author Response

The authors thank the reviewer for their comments and critiques.

In regards to the title, we respectively disagree. We mention the fact that this is only an analogous system in the abstract and deem that adequate.

The approximation used in the paper to draw the analogy does seem only to hold in the formal limit of r -> infinity. However, one finds that the approximation is within 10% of the real value by the time r ~= (3/2) M. Therefore, we claim that the approximation is quite nice even for radii within the supposed Schwarzschild radius.

The behavior of the pressure for smaller radii only determines the limiting value, while the refractive index over larger radii induces the black hole like light trajectories.

The Hawking temperature of a Solar Mass Black hole is only 10^(-8) Kelvin, so the analog black hole under consideration has a temperature of a mere 10’s of Kelvin. While this would produce some radiation, it would be so low energy as to be negligible.

We have added text in the corresponding section explaining the above.

Reviewer 2 Report

This article is well-written, it has substantial work and displayed in a clear and simple form. It should be accepted for publication.

Author Response

The authors thank the reviewer for their comments.

Reviewer 3 Report

Manuscript ID: universe-2257027

Tittle: A Cylindrical Optical-Space Black Hole Induced from High-Pressure Acoustics in a Dense Fluid

Authors: Edward Rietman, Brandon Melcher, Alexey Bobrick, Gianni Martire

The authors of this paper discuss the creation of an "optical-space cylindrical black hole" by applying a high-pressure acoustic field in a dense fluid to tune the refractive index distribution. They identify the resultant optical metric related to the refractive index function as an approximate Schwarzschild metric caused by a real black hole with a mass of about 1/1000 of the Earth's mass. The researchers explore this phenomenon and its implications from various perspectives, including comparisons to fiber optics and potential applications in optoelectronics and solar light harvesting. They also suggest that the study may have implications for quantum gravity and cosmology research. 

Although the research is interesting, it is not very convincing. Before considering the recommendation of this paper for publication, the authors should address the following questions:

1.  Eq.(8) describes the refractive index induced by the pressure wave, which is not in the same form as Eq.(4). As a result, they can only be identified when 2r>>M. However, in this limit, the metric of a black hole is the same as that of a usual star. Therefore, it is unclear what the practical application of this identification would be. Is it possible to create an optical black hole that operates at the 2r

2.  What is the purpose of studying these analog astrophysical phenomena? How could this study contribute to the fields of quantum gravity and cosmology research?

3.  What are the limitations or challenges of this research?

4. Are there any other analog astrophysical phenomena that could be studied using similar methods?

Author Response

The authors thank the reviewer for their comments and critiques.

  1. The approximation used in the paper to draw the analogy does seem only to hold in the formal limit of r -> infinity. However, one finds that the approximation is within 10% of the real value by the time r ~= (3/2) M. Therefore, we claim that the approximation is quite nice even for radii within the supposed Schwarzschild radius.

The behavior of the pressure for smaller radii only determines the limiting value, while the refractive index over larger radii induces the black hole like light trajectories.

We have added text to the corresponding section that details this.

  1. The main upshot of these analog systems is that they bring non-terrestrial and massive phenomena onto the scientists bench top. When the analogy is good, observables in the real gravitational system can be matched to those in the analog system (dark spot in the cylinder is the black hole in space). If one can develop an analogy between some classical tabletop system and a quantum gravitational system, the observations may provide clues as to how these things work in nature.
  1. What are the limitations or challenges of this research? Of this particular approach - that is the high-pressure applied to a dense fluid - the limitations, or challenges, are very few that we know of. It is a rather simple experimental setup. Using a more complex arrangement of acoustic transducers and power electronics, it may be possible for example, to create a spinning black hole. That being said however, we believe the greater challenge is not in this approach, but in understanding and exploiting other general relativistic and quantum gravity experimental platforms that can be easily constructed in a benchtop. As we review in Section 1.2 there are a very wide range of benchtop and computational analog models to draw from. Our manuscript adds to the list.
  1. We have re-emphasized the review article in the introduction. The authors also note that a number of cosmological analog models have been provided in the introduction.

Round 2

Reviewer 1 Report

The field of Black Hole analogues using laboratory experiments is a growing  and important one, but it is also distinct from Astrophysical black holes. I maintain that this distinction should be clear in the title - including the prefix "analog" to "black holes". I leave the decision to the Editors.

Reviewer 3 Report

The final part of my first question "Is it possible to create an optical black hole that operates at the 2r

I can recommend this paper for publication but I hope the authors can consider my question and give a brief discussion about this problem in this paper. 

Author Response

We apologize for a little bit of reading miscomprehension.

To build an optical analog that remains valid for radii less than r_s/4 requires two additional things.

1) Schwarzschild geometry should really not be trusted at such small radii. The presence of a singularity will not be able to be matched. So, one requires some model for a non-singular black hole.

2) More precise pressure control. In our set-up, the vibrations are applied at the boundary of the glycerin. One would need to have more control at the center to begin creating more complicated analogs. So, it may be possible, but that experimental set-up will be more complicated than our own.

We have added text below line 202 that states essentially this with a reference to a regular black hole construction.